# Parental Accompaniment in Operating Rooms Reduces Child Anxiety

**DOI:** 10.3390/healthcare11162289

**Published:** 2023-08-14

**Authors:** Harumi Ejiri, Hideto Imura, Reizo Baba, Akiko Sumi, Akiko Koga, Kaoru Kanno, Miho Kunimoto, Kayo Hayami, Teruyuki Niimi, Shuji Nomoto, Nagato Natsume

**Affiliations:** 1College of Life and Health Sciences, Chubu University, 1200 Matsumoto, Kasugai 487-8501, Aichi, Japan; babar@isc.chubu.ac.jp; 2Division of Research and Treatment for Oral and Maxillofacial Congenital Anomalies, School of Dentistry, Aichi Gakuin University, 2-11, Suemori-Dori, Chikusa-ku, Nagoya 464-8651, Aichi, Japan; h-imura@dpc.agu.ac.jp (H.I.); niimi@dpc.aichi-gakuin.ac.jp (T.N.); natsume@dpc.aichi-gakuin.ac.jp (N.N.); 3Hokkaido University Hospital, Kita14, Nishi5, Kita-ku, Sapporo 060-8648, Hokkaido, Japan; sumiakiko@huhp.hokudai.ac.jp (A.S.); kunimoto@huhp.hokudai.ac.jp (M.K.); 4Japanese Cleft Palate Foundation, 2-5 Hoocho, Chikusa-ku, Nagoya 464-0057, Aichi, Japan; 5Department of Surgery, School of Dentistry, Aichi Gakuin University, 2-11, Suemori-Dori, Chikusa-ku, Nagoya 464-8651, Aichi, Japan; snomoto@dpc.agu.ac.jp

**Keywords:** voluntary medical support, accompanying into operating rooms, anxiety

## Abstract

Background: We believe that parental presence before the induction of anesthesia for surgery among children with a cleft palate/lip would be effective in mitigating their preoperative anxiety. Objective: We assessed the states of patients with a cleft palate/lip when their parents accompanied them into operating rooms and clarified their and their parents’ cognition using a questionnaire. Methods: Data were collected via nursing observation when patients and their parents entered the operating room. Furthermore, an anonymous questionnaire was administered to patients and parents after the operation regarding their feelings about parental presence in the operating room. Results: In total, nine patients cried when they entered the surgical room. Furthermore, six patients and three parents reported preoperative anxiety. In addition, eight patients agreed that they were satisfied with the presence of their parents before induction. Conclusion: Approximately half of the patients cried. However, the presence of parents before the induction of anesthesia was effective in reducing anxiety among most patients and their parents.

## 1. Introduction

In many instances, the dental treatment of young children requires using general anesthesia [1]. However, children undergoing anesthesia is usually an anxiety-ridden time for both parents and child [2]. Induction of anesthesia can be a frightening event for children, and 60% suffer anxiety during the presurgical period [3]. Furthermore, 37% develop anxiety when undergoing surgery, and 18% develop intense anxiety upon entering the operating room [4]. Preoperative anxiety in young children undergoing surgery has been associated with a further painful postoperative recovery and a higher incidence of sleep and delirium [5]. Therefore, children’s transfer to the operating room and the smooth induction of anesthesia without heightening their anxiety is important for minimizing their perioperative distress and improving behavioral outcomes [6].

Orofacial clefts, notably cleft lip and cleft palate, are the most common craniofacial birth defects in humans, representing a substantial personal and societal burden [7]. Patients with cleft lip and cleft palate face difficulties in feeding, speech, hearing, and dental problems, and therefore require appropriate treatment by specialists. Since the provision of overseas medical support by the Japanese Cleft Lip and Palate Foundation in Ben Tre Province, Vietnam, there have been difficulties in communication with patients undergoing surgery. These difficulties stem not only from the language difference between the medical practitioners and the patients but also from the patients being children. A previous study reported that limited communication between family members and intensive care unit (ICU) staff was an important healthcare professional-related factor associated with a higher incidence of anxiety [8]. Rusinova et al. also emphasized the anxiety effect of limited communication. Several methods have been studied to reduce a child’s anxiety prior to and at induction, including premedication, music interventions, and parental presence in the operating room [9,10,11]. Furthermore, parental presence during anesthesia induction has been utilized to minimize anxiety among children, as well as to increase parental satisfaction, and avoid impeding operating room efficiency [12,13]. Previous studies have also clarified that parents of patients with cleft palatal lips present higher levels of stress in the pre-surgery period [14]. Therefore, we allowed parental presence during the induction of anesthesia to reduce anxiety in children with a cleft lip/palate and their parents in Vietnam.

This study aimed to assess the anxiety states of patients with a cleft palate/lip when their parents accompanied them into the operating room and clarify the cognition of patients and parents. To the best of our knowledge, this was the first study to assess the effect of parents accompanying their children into the operating room among patients with a cleft palate/lip in a medical volunteer situation. This study will add new insights regarding patients undergoing surgery by the overseas medical support team in Vietnam.

## 2. Materials and Methods

The study purpose, methods involved, and questionnaire data collection processes were explained to participants in detail both verbally and in writing; written consent was obtained to evaluate the anxiety states of patients based on their expression when they entered the operating room at a hospital in Vietnam. The study was performed in accordance with the principles of the Helsinki Declaration. This study was approved by the Chubu University Ethics Review Board (Approval No. 20190094).

### 2.1. Participants and Medical Support

We used a cross-sectional study design. The target population was 19 patients with a cleft lip/palate, and five parents in Ben Tre Province, Vietnam. The survey was conducted in mid-March 2020.

We provided medical support for 10 days. First, patients with a cleft lip/palate registered with the authorities for free cleft lip/palate operation by the Japanese medical support team. Second, patients underwent a preoperative medical check by the Japanese medical support team. The day before the operation, patients were hospitalized and accompanied by their parents. Patients also underwent dental procedures before entering the operating room with a dental hygienist. Subsequently, the Vietnamese and Japanese nurses guided the patients and parents to the operating room. When patients were moved to the operating bed, the accompanying parents exited the operating room. After the operation, a Japanese pediatrician, Japanese nurses, and Vietnamese nurses observed the postoperative patients in the ICU.

### 2.2. Survey

Japanese medical support nurses evaluated the anxiety states of patients based on their expression when they and their parents entered the operating room. Patients’ expressions were evaluated via an original scale: could not stop their tears, looked sad or expressionless, or smiled. Furthermore, an anonymous questionnaire was administered to the participants in Vietnamese after the operation. The questionnaire included items regarding participant characteristics such as age and gender, and five Yes/No questions regarding their feelings about parents accompanying them into the operating room: “Did you feel anxiety about the operation?”; “Were you relieved about your parents accompanying you into the operating room?”; “Did you agree for your parents to accompany you into the operating room?”; “Did you feel that your tears reduced because your parents accompanied you into the operating room?”; and “Did you feel that your fear reduced because your parents accompanied you into the operating room?” Respondents were also asked: “How did you feel when your parents accompanied you into operating room?” If the patient was too young, they answered questions on the effects of the presence of family members.

### 2.3. Analysis

The number and percentage of responses to the evaluation and all questions were ascertained by descriptive statistics used for IBM SPSS Statistics version 27. In addition, we believed that free description expresses what participants actually felt and thus was very important. It is essential for medical personnel to obtain broader knowledge about participants’ perceptions [15]. Thus, we analyzed free descriptive answers from eight participants (six parents and two patients) used for IBM text analysis for the survey (TAfS) and thematic analysis.

Textual analysis is the objective analysis of textual data using a computer through a method called text mining, which finds information from text data digitally [16]. We analyzed the keyword frequency of free descriptive data using TAfS to identify the trend in descriptive content.

In addition, we referred to the thematic analysis methods of Braun and Clarke [17], and the free descriptive answers were analyzed by experienced qualitative researchers. Thematic analysis is a method for identifying, analyzing, and reporting patterns (themes) within data. It is compatible with both essentialist and constructionist paradigms in psychology. Because of its theoretical freedom, thematic analysis provides a flexible research tool, which can potentially provide a rich, detailed, and complex account of data [17].

### 2.4. Outcomes Measures

The primary endpoint was an evaluation of anxiety in patients who underwent an operation of a cleft lip/palate before and during the induction of anesthesia and their parents. Nurses from the Japanese medical support team observed and evaluated the patients based on their expressions as they entered the operating room with their parents. Furthermore, we evaluated patients’ and parents’ anxiety and their cognition regarding accompaniment into the operating room using an original questionnaire.

## 3. Results

### 3.1. Participants

In total, 19 participants (under six years, 78.9% and over six years, 21.1%) were evaluated by Japanese nurses when they entered the operating room. Participants were 9 (47.4%) male, 10 (52.6%) female. Of these, 10 and 9 participants underwent operations for cleft lip, which included re-construction and cleft palate, respectively.

Five parents accompanied the patient to the operating room and completed a questionnaire. All five parental participants were the patients’ mothers.

### 3.2. Participants’ Anxiety States When They Entered the Operating Room

In total, 14 participants (73.7%) held onto their parents when they entered the operating room. Furthermore, nine (47.3%) could not stop their tears (Figure 1). Among them, two patients also cried severely. In addition, two patients cried violently, and five showed resistance against nurses and doctors. Five (26.3%) of the patients observed looked sad, while four (21%) patients were expressionless.

### 3.3. Cognition of the Patients and Parents

Figure 2 shows the results of the survey on patients’ cognition. In total, six (66.7%) felt anxiety regarding the operation, and eight (88.9%) were relieved that their parents accompanied them into the operating room. All patients agreed for their parents to accompany them. Also, seven patients (77.8%) reduced their fear, and six (66.7%) reduced their tears because of the presence of parents before the induction of anesthesia.

Of the five parents, three (66.7%) expressed anxiety and four (80%) agreed to accompany their child into the operating room.

In the free descriptions, text analysis revealed that the words “joy” (*n* = 5), “fear” (*n* = 4), “mother” (*n* = 3), and “accompany” (*n* = 3) frequently appeared. Thematic analysis revealed that parents felt “fear of entering the operating room.” In contrast, the patients felt feelings of “joy of parents’ accompaniment,” “fear of surgery,” and “relief from anxiety due to parents’ accompaniment.” Descriptive data of parents’ feelings of “fear of entering the operating room” included “*I feared entering into the operating room*” and “*I feared entering into the operating room. However, I was very happy as I supported my child in the operating room.*” Descriptive data of patients’ feelings of “joy of parents’ accompanying them” included “*I enjoyed being accompanied to the operating room with my mother*” and “*I was glad to be with my mother when I went into the operating room.*” Descriptive data of “fear of surgery” was “*I feared the operation.*” Descriptive data of “relief from anxiety due to parents’ accompanying” was “*I did not fear the operation as I entered the operating room with my mother.*”

## 4. Discussion

This study examined the states of 19 patients with a cleft palate/lip when their parents accompanied them into the operating room and clarified the cognition of the patients and parents. To the best of our knowledge, this was the first study to demonstrate that patients with a cleft palate/lip felt anxiety relief when their parents accompanied them into the operating room before the induction of anesthesia.

In this study, many patients under the age of six cried when they entered the operating room, suggesting that they were very anxious. In addition, it became clear that many of the subjects thought that going into the operating room with their parents was positive. Preoperative anxiety in young children undergoing surgery was associated with a further painful postoperative recovery and a higher incidence of sleep and delirium [5]. A previous Japanese study on perioperative anxiety in pediatric surgery reported that 37% of patients who underwent surgery developed anxiety. Furthermore, among children aged under one year, 61% cried or resisted in the operating room before induction [4]. We found that approximately 60% of patients aged under six years could not stop their tears when their parents accompanied them into the operating room. In addition, approximately half had preoperative anxiety. Furthermore, according to the descriptive data, most patients felt joy and anxiety relief when their parents’ accompanied them. Our results suggest that patients undergoing operations by the medical support team had a degree of anxiety similar to that found in previous studies [4]. This anxiety was considerably reduced by parental presence before anesthesia induction.

Approximately 60% of parents whose children underwent an operation felt anxiety in Japan. After the operation, 99% answered that they were happy to enter the operating room with their children [4]. Hence, it is important to inform and reassure parents whose children are undergoing a medical procedure with appropriate explanations suited to their comprehension level [18]. In this study, parents felt fear of entering the operating room; however, almost all agreed to enter. Similarly, studies regarding parental presence in the operating room during the induction of anesthesia revealed that parents had positive feelings regarding being present, even if they found the induction of anesthesia to their child as traumatizing or distressing to witness [13]. Therefore, the medical support team should appropriately explain the operation and procedure to the patient and their parents to avoid parents feeling trauma or distress.

In addition, Ismal and Mahrous [19] reported a study where parental presence was allowed during the induction of anesthesia and the mother used a scented anesthesia mask on the child. She started to encourage the child to take multiple deep breaths through the mask until they began to lose consciousness. Parents’ active participation in anesthesia induction was effective in decreasing anxiety levels among both the patients and parents. Future trials are required in our team and local teams to appropriately involve parent’s active participation, and advise the patients and parents in the operating room.

Conversely, anxiety and fear did not reduce in a few patients, even with parental presence before the induction. They may not have felt comfortable entering the room with their parents due to their parents’ increased anxiety. According to Tabaquim, parents of patients with cleft palatal lip present higher levels of stress in the pre-surgery period, including significantly unsatisfactory bodily reactions [14]. It is thus important for the medical support team to understand that parents are under high stress. However, parental presence at the induction of anesthesia should still be considered a valuable tool to improve patient and family satisfaction [12]. Thus, we should ensure that patients and parents enter the operating room together. Furthermore, we must evaluate satisfaction among both patients and parents. In addition, preoperative preparation is important to reduce anxiety for patients and parents [20]. Chan et al. found that educating patients’ parents about surgery and postoperative recovery rooms was beneficial in reducing anxiety [21]. Therefore, in overseas medical support, we should improve preoperative preparation and education for parents to reduce anxiety, and to help patients and parents understand the operation and postoperative care.

Unfortunately, the onset of the COVID-19 pandemic made it difficult for us to obtain overseas medical support and continue this study. As such, with a limited number of subjects, our findings are difficult to generalize. Other limitations of this study are its cross-sectional nature, which means we were unable to identify any variation over time. In addition, our study was not comprehensive; a comparative study with more subjects and a control group is required. Another limitation is that the anxiety states of the patients with a cleft palate/lip were evaluated on an original score. It would be useful to conduct surveys using scales such as the modified Yale Preoperative Anxiety Scale (mYPAS), the State, Visual Analogue Scale (VAS), and the Trait Anxiety Inventory (STAI). Lastly, this study was conducted through an interpreter, and participants’ detailed feelings and cultural views may hence not have been fully expressed.

## 5. Conclusions

We found that approximately 60% of patients with cleft lip/palate aged under six years could not stop their tears when their parents accompanied them into the operating room. In addition, patients had preoperative anxiety and parents felt fear of entering the operating room. Furthermore, most patients and parents felt joy and relief from anxiety when the parent also entered the operating room. These results suggested that parental presence during the induction of anesthesia reduced anxiety among patients and parents in overseas medical support in Vietnam.

## Figures and Tables

**Figure 1 healthcare-11-02289-f001:**
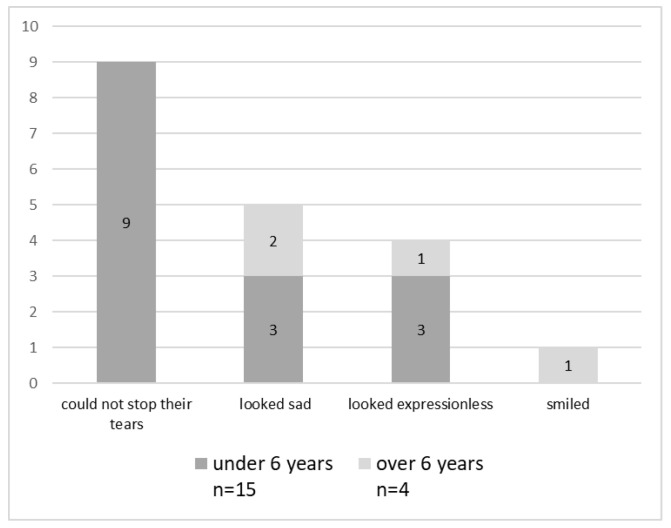
Expression of the patients when they entered operating room (*N* = 19).

**Figure 2 healthcare-11-02289-f002:**
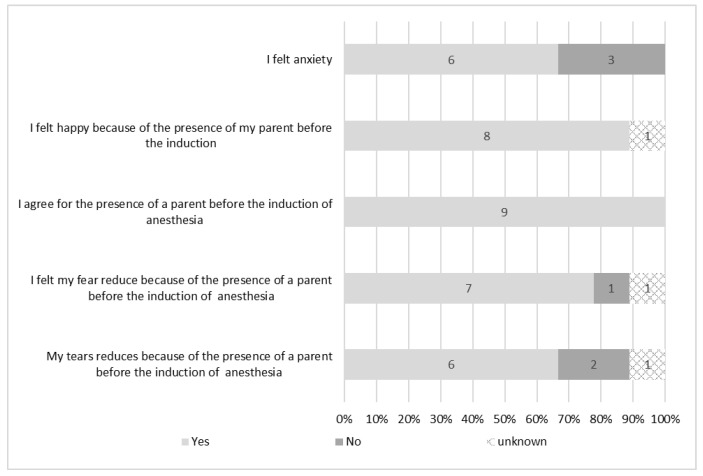
Patients’ cognition of the presence of their parents before the induction of anesthesia (*N* = 9).

## Data Availability

Not applicable.

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
