# Peer review of "Parental Accompaniment in Operating Rooms Reduces Child Anxiety"

_healthcare, 2023, doi:10.3390/healthcare11162289_

Round 1

Reviewer 1 Report

It was my pleasure to review the manuscript entitled (Children and parents' impressions on parents’ accompanying the child  into operation rooms: our experiences from a Japanese medical support team; Manuscript ID: healthcare-2471269). The study aimed to assess the anxiety of children with cleft lip and/or palate who are going to have surgical operation by the Japanese medical support team. However, it is of little significance and suffers from fundamental errors in study design and interpretation. Thus, I do not endorse its publication.

Not applicable

Author Response

Thank you for reviewing our manuscript. We have carefully considered all comments and revised our original manuscript accordingly. Here, we describe our revisions and responses to your comments.

Point1.The study aimed to assess the anxiety of children with cleft lip and/or palate who are going to have surgical operation by the Japanese medical support team. However, it is of little significance and suffers from fundamental errors in study design and interpretation. Thus, I do not endorse its publication.

Response1. To our best knowledge, no previous study evaluated parents’ accompanying their child into operating room as medical support. This was the first study to demonstrate that patients with a cleft palate/lip felt relief from their anxiety when their parent accompanied them into the operating room before the induction of anesthesia.

Although the level of evidence in this study is not high, we believe that this study will contribute to future studies related to overseas medical assistance.

Reviewer 2 Report

Authors represented an interesting  brief report a out children and parents' impressions on parents’ accompanying the child into operation rooms.

I have only minor suggestion for authors to add recommendation for future studies at  the end of the section Discussion.

Author Response

Thank you for reviewing our manuscript. We have carefully considered all comments and revised our original manuscript accordingly. Here, we describe our revisions and responses to your comments.

Point1.I have only minor suggestion for authors to add recommendation for future studies at the end of the section Discussion.

Respose1: Thank you very much for pointing this out. We have some limits and assignment for future studies. Therefore, we have added to the revised manuscript with the following.

Unfortunately, we could not continue this study and medical support owing to the coronavirus disease (COVID-19) pandemic. This made it difficult to obtain overseas medical support and to continue this study, and the number of subjects was limited. Therefore, we consider this finding difficult to generalize. This study has several limitations. First, this was a cross-sectional study. Hence, we were unable to identify any variation over time. In addition, in our study were not comprehensive study. We need a comparative study with more subjects and a control group. Second, the anxiety states of the patients with a cleft palate/lip was evaluated on an original score.  We also need more surveys using scales such as modified Yale Preoperative Anxiety Scale (mYPAS) and State, Visual Analogue Scale (VAS) and Trait Anxiety Inventory (STAI). Third, this study was conducted through an interpreter. Hence, participants’ detailed feelings and cultural views may not have been fully expressed. (p6)

Reviewer 3 Report

Dear Authors, this paper about children and parents impression on parents accompanying  the child into operation room is quite interesting but must be improved.

Please, modify the title and make it easier to read.

Abstract: describe better the materials and methods part, it is not completely understandable.

Introduction: this part is really important and in this paper in too short, please improve it adding a small chapter about overall quality of life of children undergoing general anesthesia. this paper could help: Ludovichetti FS, Zuccon A, Cantatore D, Zambon G, Girotto L, Lucchi P, Stellini E, Mazzoleni S. Early Childhood Caries and Oral Health-Related Quality of Life: Evaluation of the Effectiveness of Single-Session Therapy Under General Anesthesia. Eur J Dent. 2022 Oct 28.

Materials and methods: ok

Results: ok

Discussion:The discussion does not compare the findings of this study with existing literature or similar studies in the field. It would be beneficial to discuss how the results of this study align with or differ from previous research on preoperative anxiety in pediatric patients undergoing surgery. Including a comparative analysis would provide a broader context for the study and enhance its scientific value.

The discussion does not provide sufficient details about the methods used in the study. It would be helpful to describe the study design, sample size, data collection procedures, and statistical analysis methods employed. This information would allow readers to evaluate the study's methodology and determine the reliability and validity of the findings.

Although the discussion briefly mentions the study's limitations, it does not adequately discuss their potential impact on the results and interpretation. It is essential to address the limitations in more detail and provide insights into how these limitations might affect the generalizability and reliability of the findings. Additionally, suggestions for future research to overcome these limitations could be included to encourage further investigation in the field.

Author Response

Thank you for reviewing our manuscript. We have carefully considered all comments and revised our original manuscript accordingly. Here, we describe our revisions and responses to your comments.

Point1. Dear Authors, this paper about children and parents impression on parents accompanying the child into operation room is quite interesting but must be improved. Please, modify the title and make it easier to read.

Respose1: Thank you very much for pointing this out. We revised study title.

“Parental accompanying into the operation rooms reduces children’s anxiety”

Point2. Abstract: describe better the materials and methods part, it is not completely understandable.

Respose2: Thank you very much for pointing this out. We revised abstract with the following.

Abstract: Background: We believe that paternal presence before the induction of anesthesia for surgery among children with a cleft palate/lip would be effective in mitigating their preoperative anxiety. Objective: We assessed the states of patients with a cleft palate/lip when their parents accompanied them into operation room and clarified the cognition of the patients and parents using the questionnaire. Methods: Data were collected via nursing observation when patients and their parents entered the operating room. Furthermore, an anonymous questionnaire was administered to the patients and parents after operation. The questionnaire enquired profile questions and five questions regarding their parents accompanying them into operation room. Results: In total, nine cried when they entered the surgical room. Furthermore, six patients and three parents reported preoperative anxiety. In addition, eight patients agreed that they were satisfied with the presence of their parents before induction. Conclusion: Approximately half of the patients cried. However, the presence of parents before the induction of anesthesia was effective in reducing anxiety among most patients and their parents.

Point3. Introduction: this part is really important and in this paper in too short, please improve it adding a small chapter about overall quality of life of children undergoing general anesthesia. this paper could help: Ludovichetti FS, Zuccon A, Cantatore D, Zambon G, Girotto L, Lucchi P, Stellini E, Mazzoleni S. Early Childhood Caries and Oral Health-Related Quality of Life: Evaluation of the Effectiveness of Single-Session Therapy Under General Anesthesia. Eur J Dent. 2022 Oct 28.

Respose3: Thank you very much for pointing this out. And I would like to thank you for introducing me to a reference study. In our manuscript, there were many deficiencies in the introduction. Therefore, we have added to the revised manuscript with the following.

Introduction

Child is uncooperative and it is not possible to safety perform dental procedures, the dental treatment of young children single-session therapy using general anesthesia improve their symptom and quality of life [1]. Treatment and procedures using general anesthesia are also necessary to improve the quality of life of pediatric patients. However, general anesthesia present potential risks to the general health and mental situation of patients. Children undergoing anesthesia is usually an anxiety-ridden time for both the parents and child [2]. Induction of anesthesia can be a frightening event for children, and 60% suffer anxiety during the presurgical period [3]. Furthermore, 37% develop anxiety when undergoing surgery, and 18% develop intense anxiety upon entering the operating room [4]. Preoperative anxiety in young children undergoing surgery has been associated with a further painful postoperative recovery and higher incidence of sleep and delirium [5]. Therefore, children’s transfer to the operating room and the smooth induction of anesthesia without heightening their anxiety is important for minimizing their perioperative distress and improving behavioral outcomes [6].

Orofacial clefts, notably cleft lip and cleft palate, are the most common craniofacial birth defects in humans and represent a substantial personal and societal burden [7]. Patient with cleft lip and cleft palate, difficulties feeding and speech, hearing and dental problems. Therefore, the patient with cleft lip/ palate needs appropriate treatment by specialists. Since the beginning of overseas medical support, by the Japanese Cleft Lip and Palate Foundation, in the Ben Tre Province, Vietnam, there have been difficulties in communication with patients undergoing surgery. Not only the language difference due to the difference in the country of the medical practitioner and the patient, but also it is difficult to understand the treatment because the patient is a child. A previous study reported that limited communication between family members and intensive care unit (ICU) staff was an important healthcare professional-related factor associated with a higher incidence of anxiety [8]. Rusinova et al. emphasized the anxiety effect of limited communication. Several methods have been studied to reduce a child’s anxiety prior to and at induction, including premedication, music interventions, parental presence into operating room [9-11]. Furthermore, parental presence during anesthesia induction has been utilized to minimize anxiety among children [2,3]. In addition, parental presence at induction of anesthesia to increase parental satisfaction and parents felt positive feeling, and may not impede operating room efficiency [12,13]. Also, previously study clarified that the parents of the patients with cleft palatal lip, in surgical situation, present higher levels of stress in the period of pre-surgery [14]. We needed parents care of the patients with cleft palate and cleft lip. Therefore, we allowed parental presence during the induction of anesthesia to reduce anxiety in children with a cleft lip/palate and their parents in Vietnam.

This study aimed to assessed the anxiety states of the patients with a cleft palate/lip when their parents accompanied them into operating room and clarify the cognition of the patients and parents. To our best knowledge, this was the first study to assess the effect of parents accompanying their children into the operating room among patients with a cleft palate/lip in a medical volunteer situation. This study will add new insights into patients undergoing surgery by the overseas medical support team in Vietnam.

Point4. Materials and methods: ok

Respose4: Thank you very much for comment.

Point5. Results: ok

Respose5: Thank you very much for comment.

Point6. Discussion: The discussion does not compare the findings of this study with existing literature or similar studies in the field. It would be beneficial to discuss how the results of this study align with or differ from previous research on preoperative anxiety in pediatric patients undergoing surgery. Including a comparative analysis would provide a broader context for the study and enhance its scientific value.

Respose6: Thank you very much for pointing this out. We have added to the revised discussion of manuscript with the following.

In this study, many patients under the age of 6 cried when they entered the operating room, suggesting that they were very anxious. In addition, it became clear that many of the subjects thought that going into the operating room with their parents was positive. Preoperative anxiety in young children undergoing surgery was associated with a further painful postoperative recovery and higher incidence of sleep and delirium [5]. A previous Japanese study on perioperative anxiety in pediatric surgery reported that 37% patients who underwent surgery developed anxiety. Furthermore, among children aged under one year, 61% cried or resisted in the operating room before induction [4]. We found that approximately 60% of patients aged under six years could not their stop tears when their parents accompanied them into the operating room. In addition, approximately half had preoperative anxiety. Furthermore, most patients felt joy and relief of anxiety by descriptive data when their parents’ accompanied them. Our results suggested that patients who underwent operation by the medical support team had a degree of anxiety similar to the previous study [4]. Hence, paternal presence before the induction of anesthesia was suggested. Furthermore, parental presence before induction among children with a cleft lip/palate reduced their anxiety in Vietnam.

Approximately 60% of the parents whose children underwent an operation felt anxiety in Japan. After the operation, 99% answered that they were happy to enter the operating room with their children [4]. Hence, it is important to inform and reassure the parents, whose children are undergoing a medical procedure, with appropriate explanation suited to their comprehension level [15]. In this study, parents felt fear of entering the operation room; however, almost all agreed to enter. Studies of parental surveys regarding parental satisfaction with present in the operating room during the induction of anesthesia, parents felt positive feelings regarding having been present, perception of induction as traumatizing or distressing to witness was revealed [13]. Therefore, the medical support team should appropriately explain the operation and procedure to the patient with a cleft palate/ lip and their parents to avoid parents feel traumatizing or distressing.

In addition, Ismal and Mahrous [16] reported a study where parental presence was allowed during induction of anesthesia and the mother used a scented anesthesia mask on the child. She started to encourage for child to take multiple deep breaths through the mask until they began to lose consciousness. Parents’ active participation in anesthesia induction was effective in decreasing anxiety levels among both the patients and parents. Future trials are required in our team and local teams to appropriately involve parent’s active participation and advise the patients and parents in the operating room.     

Conversely, anxiety and fear did not reduce in few patients, even with parental presence before the induction. They may not have felt comfortable entering the room with their parents due to their parents’ increased anxiety. Tabaquim clarified that the parents of the patients with cleft palatal lip, in surgical situation, present higher levels of stress in the period of pre-surgery, with modified quality in the autonomic aspects of the organism, besides bodily significantly unsatisfactory reactions [14]. It is important for medical support team to understand that parents are under high stress. However, parental presence at induction of anesthesia should still be considered as a valuable tool to improve patient and family satisfaction [12]. Thus, we should ensure that patients and parents enter the operation room together. Furthermore, we must evaluate satisfaction among both patients and parents. In addition, preoperative preparation is important to reduce anxiety for patients and parents [17]. Chan et al. found that educating patients' parents about surgery and postoperative recovery rooms was beneficial in reducing anxiety [18]. Therefore, in the overseas medical support, we should improve preoperative preparation and education for parents to reduce anxiety, and to help the patients and parents understand the operation and postoperative care.(p5)

Point7. The discussion does not provide sufficient details about the methods used in the study. It would be helpful to describe the study design, sample size, data collection procedures, and statistical analysis methods employed. This information would allow readers to evaluate the study's methodology and determine the reliability and validity of the findings.

Respose7: Thank you very much for pointing this out. We have added to the revised discussion of manuscript with the following.

This study examined the states of the 19 patients with a cleft palate/lip when their parents accompanied them into the operating room and clarified the cognition of the 9 patients and 2 parents. The number and percentage of responses to all the evaluation and questions were ascertained by descriptive statistics. Furthermore, free descriptive answers from eight participants (six parents and two patients) were used for text analysis. Similar descriptions were categorized similar statements to understand the content.(p5)

Point8. Although the discussion briefly mentions the study's limitations, it does not adequately discuss their potential impact on the results and interpretation. It is essential to address the limitations in more detail and provide insights into how these limitations might affect the generalizability and reliability of the findings. Additionally, suggestions for future research to overcome these limitations could be included to encourage further investigation in the field.

Respose8: Thank you very much for pointing this out. We have added to the revised discussion of manuscript with the following.

Unfortunately, we could not continue this study and medical support owing to the coronavirus disease (COVID-19) pandemic. This made it difficult to obtain overseas medical support and to continue this study, and the number of subjects was limited. Therefore, we consider this finding difficult to generalize. This study has several limitations. First, this was a cross-sectional study. Hence, we were unable to identify any variation over time. In addition, in our study were not comprehensive study. We need a comparative study with more subjects and a control group. Second, the anxiety states of the patients with a cleft palate/lip was evaluated on an original score.  We also need more surveys using scales such as modified Yale Preoperative Anxiety Scale (mYPAS) and State, Visual Analogue Scale (VAS) and Trait Anxiety Inventory (STAI). Third, this study was conducted through an interpreter. Hence, participants’ detailed feelings and cultural views may not have been fully expressed. (p6)

Round 2

Reviewer 3 Report

Ok

Author Response

Response to Academic Editor Comments

The paper is included in qualitative research.
The paper should be followed by COREQ: Consolidated criteria for reporting qualitative research and SRQR: Standards for reporting qualitative research.
The authors should add more important parts.
The paper needs to be revised and add the checklist as a supplemental file.

Response:

Thank you very much for pointing this out.  We have added an additional description of the qualitative research you pointed out. However, COREQ is a guideline used for qualitative research of interview studies. Therefore, please understand that there are parts that are not suitable for this research.

2.3. Analysis

The number and percentage of responses to the evaluation and all questions were ascertained by descriptive statistics used for IBM SPSS Statistics version 27. In addition, we believed that free description expresses what participants actually felt and thus was very important. It is essential for medical personnel to obtain broader knowledge about participants' perceptions [15]. Thus, we analyzed free descriptive answers from eight participants (six parents and two patients) used for IBM text analysis for the survey (TAfS) and thematic analysis.

Textual analysis is the objective analysis of textual data using a computer through a method called text mining, which finds information from text data digitally [16]. We analysed keyword frequency of free descriptive data using TAfS to identify the trend in descriptive content.

In addition, we referred to the thematic analysis methods of Braun and Clarke [17], and the free descriptive answers were analyzed by experienced qualitative researchers. Thematic analysis is a method for identifying, analyzing and reporting patterns (themes) within data. It is compatible with both essentialist and constructionist paradigms in psychology. Because of its theoretical freedom, thematic analysis provides a flexible research tool, which can potentially provide a rich, detailed, and complex account of data [17].
